# Artificial intelligence-designed single molar dental prostheses: A protocol of prospective experimental study

Reinhard Chun Wang Chau[1], Ming Chong[1], Khaing Myat Thu[1], Nate Sing Po Chu[1], Mohamad Koohi-Moghadam[1], Richard Tai-Chiu Hsung[2], Colman McGrath[1], Walter Yu Hang Lam[1]*

**1** Faculty of Dentistry, The University of Hong Kong, Hong Kong Special Administrative Region, Hong Kong, People's Republic of China, **2** Department of Computer Science, Chu Hai College of Higher Education, Hong Kong Special Administrative Region, Hong Kong, People's Republic of China

* retlaw@hku.hk

## Abstract

### Background

Dental prostheses, which aim to replace missing teeth and to restore patients' appearance and oral functions, should be biomimetic and thus adopt the occlusal morphology and three-dimensional (3D) position of healthy natural teeth. Since the teeth of an individual subject are controlled by the same set of genes (genotype) and are exposed to mostly identical oral environment (phenotype), the occlusal morphology and 3D position of teeth of an individual patient are inter-related. It is hypothesized that artificial intelligence (AI) can automate the design of single-tooth dental prostheses after learning the features of the remaining dentition.

### Materials and methods

This article describes the protocol of a prospective experimental study, which aims to train and to validate the AI system for design of single molar dental prostheses. Maxillary and mandibular dentate teeth models will be collected and digitized from at least 250 volunteers. The (**original**) digitized maxillary teeth models will be duplicated and **processed** by removal of right maxillary first molars (FDI tooth 16). Teeth models will be randomly divided into **training** and **validation** sets. At least *200* training sets of the original and the processed digitalized teeth models will be input into 3D Generative Adversarial Network (GAN) for training. Among the validation sets, tooth 16 will be **generated by AI** on *50* processed models and the morphology and 3D position of AI-generated tooth will be compared to that of the natural tooth in the original maxillary teeth model. The use of different GAN algorithms and the need of antagonist mandibular teeth model will be investigated. Results will be reported following the CONSORT-AI.

**Data Availability Statement:** No datasets were generated or analysed during the current study. All relevant data from this study will be made available upon study completion.

**Funding:** This research is sponsored by the General Research Fund, University Grants Committee, Hong Kong (Project number: 17126021).

**Competing interests:** There are no conflicts of interest to declare and no financial interest to report.

# Introduction

## Background

**Missing teeth and its consequences.** Missing teeth, either congenital or consequent to dental diseases or trauma, is commonly seen and deteriorates patients' health, well-being, and quality-of-life [1, 2]. Epidemiological studies have projected a trend of increasing missing teeth and deterioration of oral health due to the aging [3, 4] of the society, and efforts are needed to prepare for the spike of needs in health resources.

**Biomimetic dental prostheses.** By replacing the missing teeth, dental prostheses aim to restore appearance and oral functions of a patient, thus restore patient's quality of life. The prevalence of single tooth loss is the most observed pattern of tooth loss. For the best prosthodontic prognosis and restoration of functions, dental prostheses should be biomimetic [5, 6] and thus adopt the occlusal morphology and 3D position of healthy natural teeth. These features sustain the functional equilibrium of dentition such as the bucco-lingual position as well as the occlusal relationships, and any disturbance to this equilibrium may result in pathology of the teeth and thus oral dysfunctions. It has always been a challenge to reconstruct original morphology on the dental prostheses, and treatment outcome tend to vary, subjecting to errors and failure [7]. Thus, a more personalized approach in designing prostheses is needed [8, 9].

**Limitation of Computer-Assisted Design (CAD).** Despite the wide application of CAD software, the use of virtual patient models and virtual articulators [10–12], as well as the pre-formed tooth morphology libraries facilitates designing dental prostheses over a shorter period, considerable time is still required to fit the dental prostheses into patients' occlusion clinically [13, 14]. The tooth morphology generated by these libraries do not fits in the individual patients. The duplication of the original tooth or "flipped" of the contralateral tooth, if present, requires the correct 3D tooth position, otherwise occlusal interferences will still result and requires significant clinical adjustments. Moreover, the inputs of dentists and dental technicians are still necessary, and the design process can still be lengthy, regardless of the approach taken for impression/scanning and fabrication process [15].

**Artificial Intelligence (AI) as solution.** The use of AI and dental robots are considered as the trend of automated digital dentistry that may reduce the manpower costs, and AI models have shown the potential of being a reliable tool on various dental treatment process [16, 17]. AI is defined as "the capability of a device to perform functions that are normally associated with human intelligence such as reasoning, learning, and self-improvement" [18]. With adequate trainings, AI may perform tasks that in the past require human intelligence, and in connection with this, AI may help designing dental prostheses. Since the teeth of an individual subject are controlled by the same genes (genotype) [19], and all teeth are exposed to mostly identical oral environment (phenotype), the occlusal morphologies and 3D positions of a subject's teeth are inter-related. By taking reference to the natural dentition, the prostheses in questions should be able to retain the original functionality and to have no interference to the jaw movements. It is hypothesized that using a 3D Generative Adversarial Network (GAN) [20, 21], AI can learn the morphology and positional relationship between teeth in an individual subject and automate the design of biomimetic single-tooth dental prostheses from the features of the remaining dentition after sufficient trainings.

**Generative adversarial networks.** The AI trainings in dentistry are mostly on the 2D radiography and 3D tomography [22, 23]. To the best knowledge of the authors, there is no clinical study on the AI learning of 3D teeth models. Four GAN approaches, namely voxel-based, view-based, point-based, and fusion methods, are currently used to train the 3D models [24] and this study investigates if there are any differences in these approaches in the AI training for 3D teeth models.

**Antagonist teeth models.** Regarding the prediction of occlusal morphology, studies have shown that the presence of antagonist teeth models helps in setting the limit of occlusal morphology [25]. It is unknown if the same is applied to the AI learning and predicting of the occlusal morphology.

## Objectives

- To compare the occlusal morphology and 3D position of the AI-generated single-tooth dental prostheses with reference to healthy natural tooth.

   a. To determine the appropriate AI deep-learning methods/algorithms that should be used in interpreting and learning of the features of 3D teeth models.

   b. To determine if the antagonist (mandibular) teeth models should be used in the AI training.

## Study protocol

### Materials and methods

**Collection of teeth models.** This study is designed in accordance with the Standard Protocol Items: Recommendations for Interventional Trials (SPIRIT) guidelines [26] (Fig 1).

| | Study Period | | | | | | | |
|---|---|---|---|---|---|---|---|---|
| | Enrolment | Allocation | Intervention Period | | | | | Close-out |
| TIMEPOINT** | -$t_1$ | 0 | $t_1$ | $t_2$ | $t_3$ | $t_4$ | $t_5$ | $t_6$ |
| **ENROLMENT:** | | | | | | | | |
| **Eligibility screen** | X | | | | | | | |
| **Informed consent** | X | | | | | | | |
| *Digitizing the models* | ●——————● | | | | | | | |
| *Allocation to training group or validation group* | | X | | | | | | |
| **INTERVENTIONS:** | | | | | | | | |
| *Computation & AI Training* | | | ●————————————● | | | | | |
| **ASSESSMENTS:** | | | | | | | | |
| *Matching of tooth 16 to its respective arch* | | | | ●——————————● | | | | X |
| *Geometric morphology* | | | | ●——————————● | | | | X |
| *Functional occlusal morphology* | | | | ●——————————● | | | | X |
| *3D position* | | | | ●——————————● | | | | X |

**Fig 1. SPIRIT diagram showing the schedule of enrollment, allocation, interventions, and assessments of this study.** $-t_1$ = baseline assessment (before randomization), 0 ($t_0$) = randomization, $t_1$ = after randomization, $t_2$ = 25% of collected models learnt by AI, $t_3$ = 50% of collected models learnt by AI, $t_4$ = 75% of collected models learnt by AI, $t_5$ = 100% of collected models learnt by AI, $t_6$ = the whole study is completed.

Maxillary and mandibular dentate teeth models will be collected from at least 250 volunteers [27], consisting of patients attending the University teaching hospital (Prince Phillip Dental Hospital) as well as the students of the University of Hong Kong. Participants will be screened and selected according to the following inclusion and exclusion criteria:

**Inclusion criteria.    Subject level**:

- Subjects with more than 12 functional occlusal units and stable maximal intercuspal position (MIP) [28].

- Subjects who have the right maxillary first molar tooth.

- Subjects who did not undergo orthodontic treatment and/or did not have teeth that rotated more than 45 degrees and/or displaced more than 1.5 mm [29].

**Tooth level**:

- Sound tooth or tooth with single surface restorations that do not grossly alter its morphology.

**Exclusion criteria.    Subject level**:

- Subjects with periodontal disease (Basic Periodontal Examination BPE Score 3) whereby there may be pathological tooth migration and alteration of occlusal plane.

- Subjects who are under the age of 18 and unable to give consent.

**Tooth level**:

- Teeth with extensive (two or more surfaces) restorations that affect the morphology.

- Teeth that have pathological tooth movements such as fremitus, drifting, and overeruption.

Subjects will be first screened for eligibility at the subject-level. Individual teeth in these recruited subjects will then be assessed for eligibility at tooth-level.

## Preparation of teeth models

Teeth models will be collected and digitized using laboratory scanner (Trios D2000; 3Shape, Denmark) into Standard Tessellation Language (STL) format by one operator and the quality will be checked by another operator.

Digitized models in STL format will be converted into Polygon File Format (PYL) using MeshLab (v2021.07, Visual Computing Lab of the ISTI-CNR) [30]. The converted files will be duplicated into two identical sets, with one set of files with **original** maxillary arch and one set of files with the right maxillary first molar (FDI World Dental Federation Notation 16) removed from the arch using MeshLab to simulate a missing tooth i.e. **processed** (Fig 2).

Collected teeth models will be randomly divided into **training** and **validation** sets.

At least two-hundred **training** sets including the **original** and the **processed** models will be input into the AI system to learn the relationship between individual teeth (tooth 16) and rest of the dentition via a 3D Generative Adversarial Network (GAN) designed for this study (Fig 3). Data will be computed by the High-Performance Computer provided by the Information Technology Services (ITS), The University of Hong Kong (HKU), which is capable of conducting data intensive and accelerated computing using a high-performance computing cluster with an aggregating processing power of 804 TFlop/s (Fig 4).

**Deep-learning methods.**    The following deep-learning methods/algorithms will be adopted for the training process:

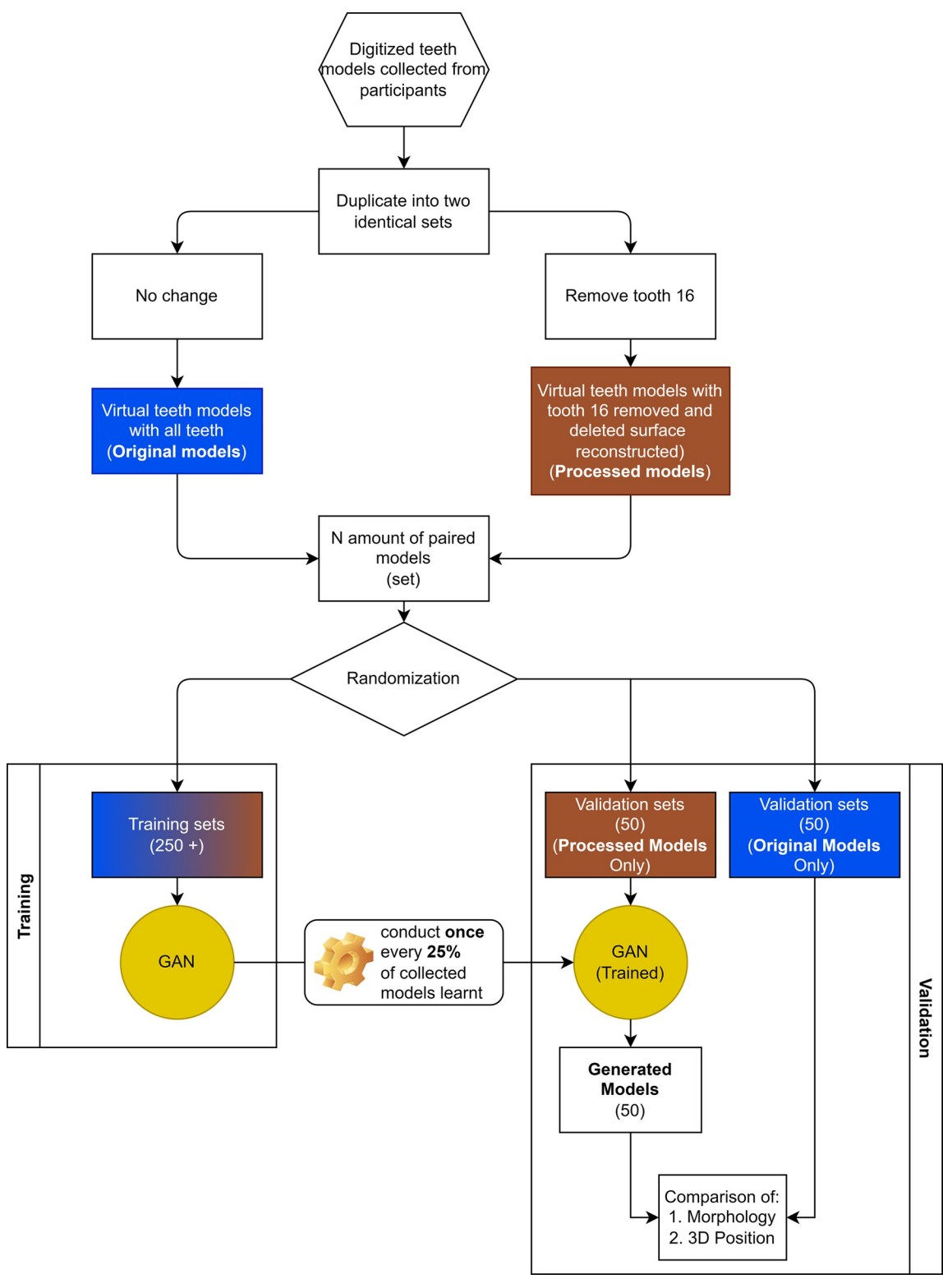

**Fig 2. General flow of processing collected data.**

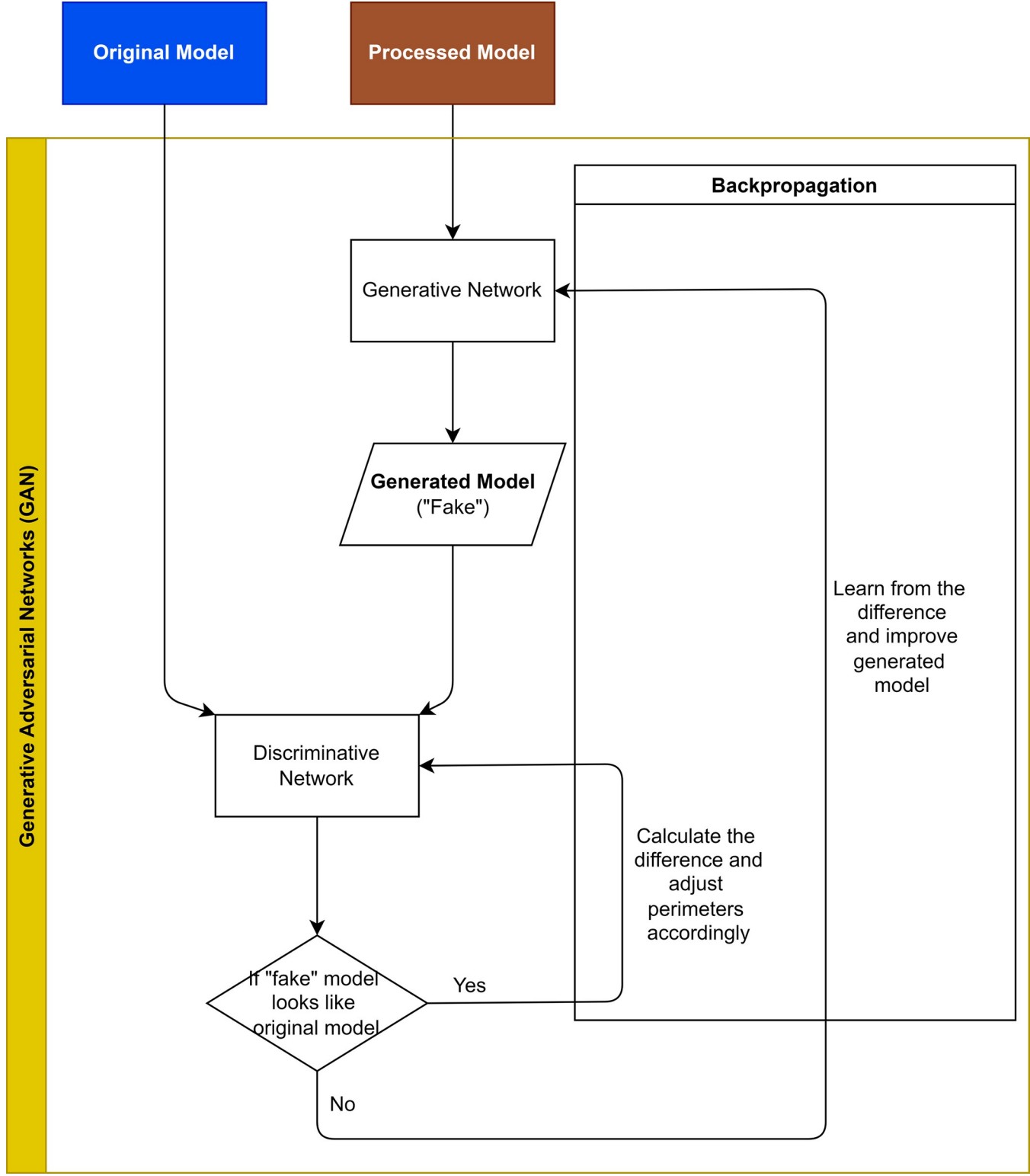

**Fig 3. Brief architecture of the Generative Adversarial Network (GAN) of this study.** Original model: digitized maxillary teeth model collected from participants; processed model: teeth model with tooth 16 removed; Generated model: teeth model with AI-generated tooth 16.

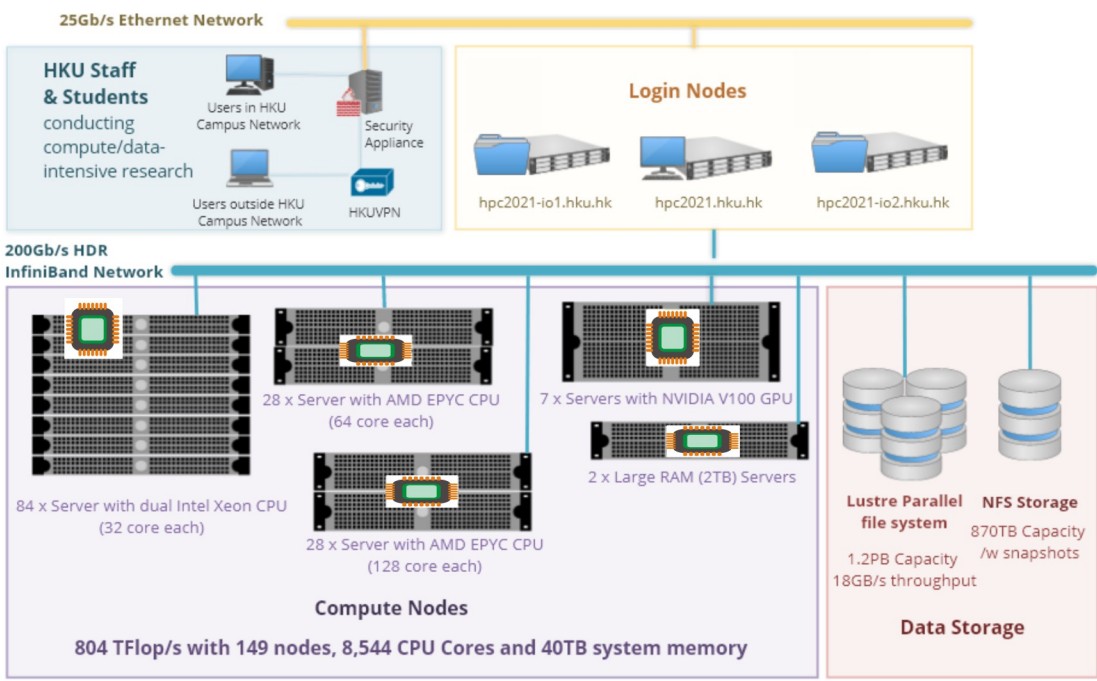

**Fig 4. System specification of high-performance computer.** Republished from [31] under a CC BY license, with permission from ITS, HKU, original copyright 2022.

**Voxel-based:** This method first partitions the 3D model into regular cubes and then inputs into the neural network (Fig 5A). In this study, the 3D vertices are sampled into 64x64x64 voxels. Example of this algorithm is VoxNet, an architecture that integrate a volumetric occupancy grid representation with a supervised 3D Convolutional Neural Network (3D CNN) [32].

**View-based:** This method uses multiple 2D image views of a 3D model for its alignment and orientation (Fig 5B). In this study, 2D image views of resolution 1920x1200 pixels are aggregated from a loop around the 3D models and 2D deep learning framework is applied to them. Example of this algorithm is Group-view convolutional neural networks (GVCNN), which is composed of a hierarchical view-group-shape architecture for 3D-shape representation [33].

**Point-based:** This method uses point-cloud to represent the 3D models (Fig 5C). A point-cloud is a set of data points in space. Each point has its set of $x$-, $y$- and $z$-coordinates. Point-clouds are generally produced by 3D scanners, and they measure many points on the external surfaces of objects around them. PointNet++ is an example of such algorithm and it functions by applying PointNet, a deep-learning method on point sets, on a nested partitioning of the inputting point sets [34].

**Fusion methods:** This method learns on multiple types of data (multi-view, point-cloud and voxel) and fuses their features. Example of this algorithm is FusionNet, which uses the volumetric grid and multi-view for classification, combining several advances in machine learning of recent years with a summation-based skip connection [35].

**Presence of antagonist teeth models.** The presence of antagonist teeth models are currently essential to the CAD design and the fabrication of the dental prostheses. These

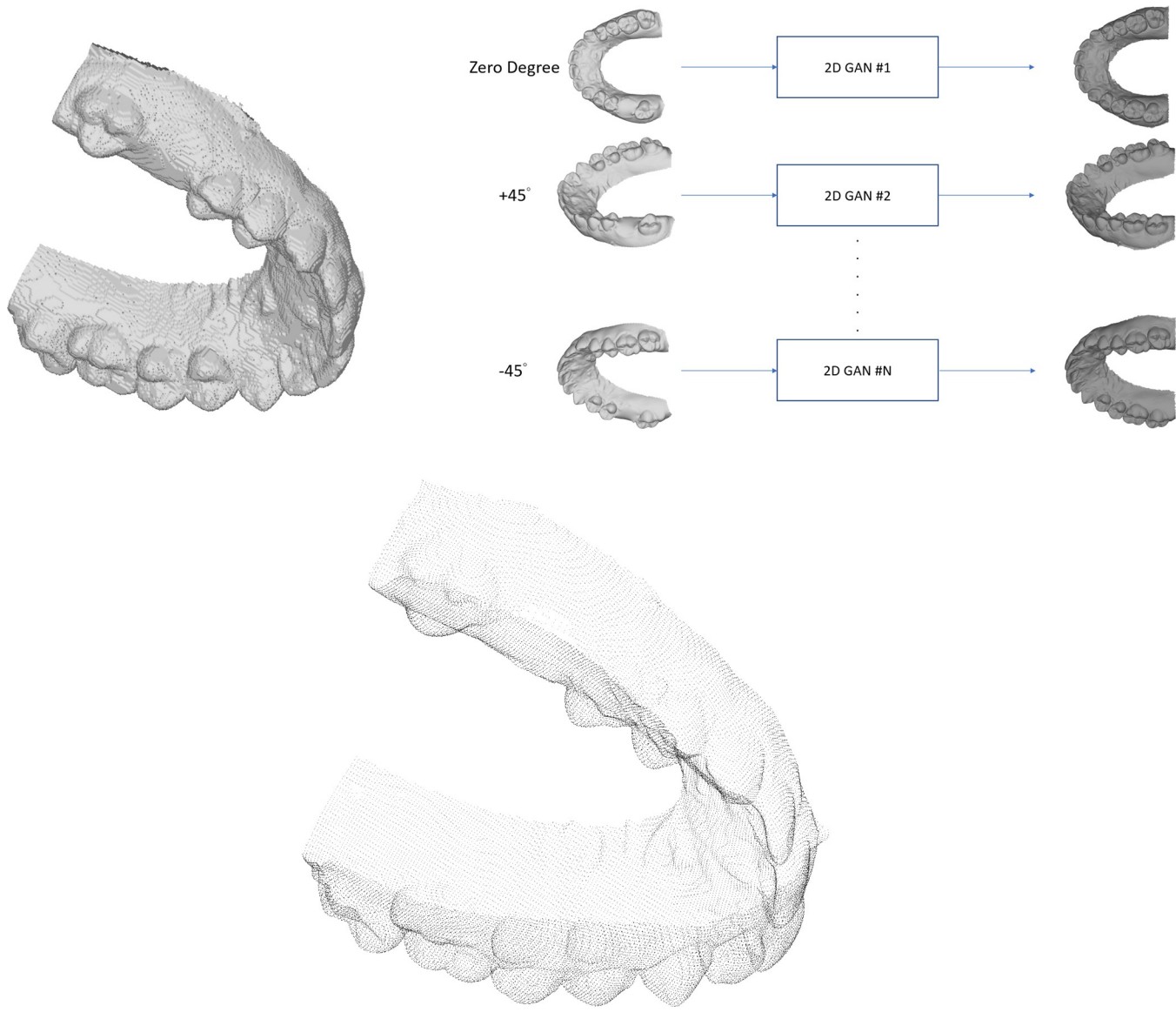

**Fig 5.** A. Illustration of voxel-based method. B. Illustration of view-based method. C. Illustration of point-based method.

antagonist models provide the information of occlusal contacts and limits the cuspal height of occlusal morphology [25]. It is unknown if the same is applied to the AI learning and predicting of the occlusal morphology. Features of the remaining teeth of an arch may be sufficient for the prediction of single missing tooth. Maxillary teeth models with and without the antagonist mandibular teeth models will be used to train the AI system (Fig 6A–6C). Additional mandibular teeth model as well as the MIP maxillomandibular relationship will be input to the AI system.

**Validation of AI system.** Separate (50) sets of the original and the processed maxillary teeth models will be serve as validation sets. The effectiveness of the four deep learning algorithms in predicting the occlusal morphology and 3D position of single missing tooth (tooth 16) on the processed models will be evaluated and compared to the natural tooth 16 on the original models.

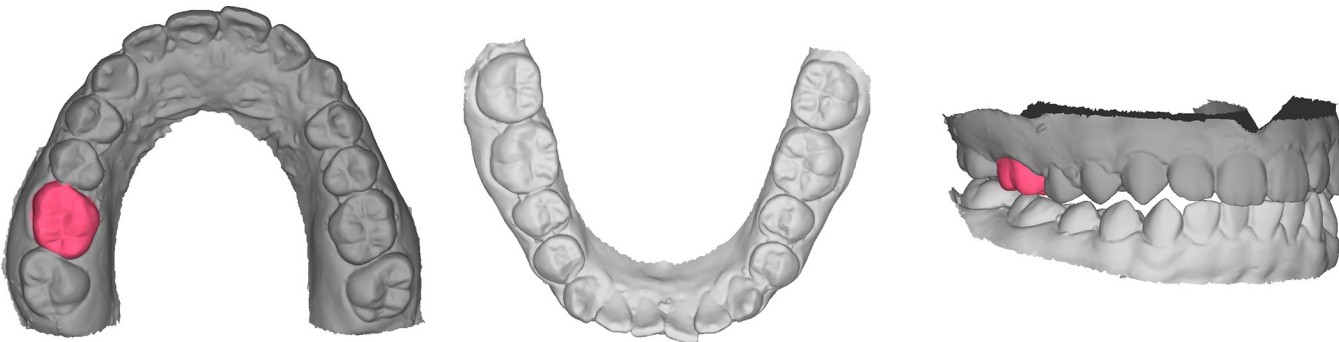

**Fig 6.** A & B: Example of a maxillary teeth model (left) and a mandibular teeth model (right). C: Demonstration of maxillary teeth model and its antagonist mandibular teeth model at maximal intercuspal position.

After determining the appropriate GAN algorithm, the need of antagonist mandibular teeth models in AI predicting the occlusal morphology and 3D position of single missing maxillary tooth will be evaluated.

## Outcome measurement

The progress of the AI training will be evaluated with quartiles analysis. The following indicators of AI-designed prostheses will be examined at least 4 times during the training procedure, when 25%, 50%, 75% and 100% of collected teeth models have been learned respectively by the AI system. The morphology, i.e., geometric morphometric, and 3D positions, of tooth 16 designed by the trained AI system will be assessed by comparing to the corresponding natural teeth among the validation sets.

**a) Occlusal morphology. 1. Matching of trimmed tooth 16 to its respective arch**–The processed models and their trimmed original tooth 16 of the validation sets will be anonymized. The AI generated tooth 16 on these processed models will be used as a guide to match the trimmed original tooth 16 to its corresponding arch. The individual trimmed original tooth 16 will be best fit superimposed to the AI generated tooth 16 on these processed models. Hausdorff distance (HD) showing the degree of match will be obtained and the trimmed original tooth with the least HD will be regarded as best fitting and will be matched to this processed model [36]. The percentage of correct matching will be reported.

**2. The geometric morphometric of AI generated tooth**–the cusp tips (highest point), fossae (lowest point) and cuspal inclines (joining the cusp and fossa) of the individual cusps will be compared to the original tooth 16 by best-fit superimposition and their differences will be analyzed (Fig 7A).

**3. Functional occlusal morphology**–the maxillary and mandibular teeth models will be mounted to a virtual articulator (exocad; DentalCAD, exocad) at average setting and the whole system is instructed to perform simulated jaw movements to determine if there is any occlusal interference [9, 37].

**b) 3D position.** The center (and its coordinate) of the AI generated tooth and the original tooth 16 will be determined by the software (Fig 7B). Using the adjacent teeth as reference area for best-fit superimposition, the buccal and lingual contours of the AI generated tooth and the original tooth will be compared, and their differences will be observed both visually as well as in HD.

## Sample size determination

Each sample, training or validation set, of this study refers to the digitalized tooth model(s) collected from one participant. In a systematic review of the sample size determination for

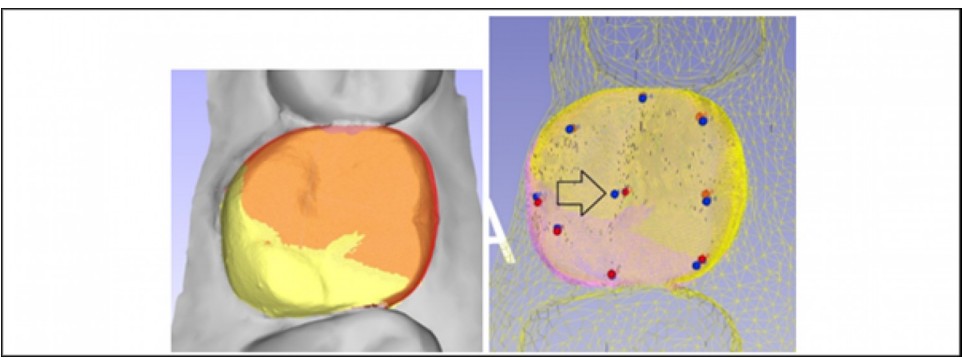

**Fig 7.** A & B: Superimposition of teeth for comparison (left). Measurement of the geometric morphology and 3D position by locating the anatomical landmarks of a tooth such as cusp tips and fossae as well as the center of a tooth (arrowed) respectively (right).

medical AI studies, only 4 studies provided rationale of sample size calculation [38], and only one study is about CNN and 3D images [39]. The recommended sample size for CNN in the medical imaging 3D computed tomography has been found to be 1000. From its Receiver Operating Characteristic (ROC) curve, the accuracy of AI system starts to reach plateau when the sample size reaches 200 and become saturated when reach 1000. Therefore, 200 subjects will be recruited as the minimal working sample size for training the AI system. More subjects that are eligible will be recruited for AI training if possible. Additional 50 subjects will be recruited for the validation sets.

## Ethics and dissemination

**Approval.**   The study will be conducted according to the Declaration of Helsinki and the research team will act in accordance with ICH GCP guidelines, local regulations and Hospital Authority Hong Kong and the University of Hong Kong policies.

The study protocol is approved by the Institutional Review Board of the University of Hong Kong/ Hospital Authority Hong Kong West Cluster (HKU/HA HKW IRB), Hong Kong Special Administrative Region, People's Republic of China (Reference Number: UW 20–848). The protocol of this study can be found at ClinicalTrials.gov (ClinicalTrials.gov ID: NCT05056948).

**Consent.**   Written consents will be obtained by research assistants and interpreters will be provided as needed.

**Confidentiality.**   All data will be kept and reviewed by unblended member to check quality. Only the data that are related to the study outcomes will be disclosed. All the data will be kept for another two years after completion of study. After that, data will be destroyed completely. Only the primary investigator will have access to the personal data during and after the study.

**Publication policy.**   Findings and data will be published in international peer reviewed journals and at similar international conferences.

## Discussion

This is a prospective experimental study regarding artificial intelligence-designed dental prostheses, as well as one of the first few projects related to applying AI in Prosthodontics for

replacement of single missing tooth. Its results may be proved crucial in future research into extended AI involvement in dentistry such as the use of specific GAN algorithm and the need of antagonistic teeth models may be used to further improve AI system for other prosthodontic procedures.

It is also yet an uncertainty on wide application of AI in various dental treatments due to the difficulties, including but not limited to standardization, repeatability, robustness and technical issues. The checklist of CONSORT-AI will be completed as supplement data when reporting the result of this study.

With sufficient resources devoted to the training of AI, like providing ample learning samples until the system is saturated and is no longer showing improvement, the outcome of the study may be different from the preliminary results. Receiver Operating Characteristic (ROC) curves will be plotted to determine the saturation/plateau of this AI system. Further studies will be conducted to examine the generalizability (external validity) of this AI system by including subjects who have received orthodontic treatments and/or with history of periodontal diseases in the validation sets. The limitation of this AI system includes teeth movement within periodontal ligament was not considered.

## Supporting information

**S1 Checklist. SPIRIT 2013 checklist: Recommended items to address in a clinical trial protocol and related documents**∗**.**
(DOC)

**S1 File.**
(DOCX)

**S2 File. Request for permission to publish content under CC-BY license.**
(PDF)

**S1 Protocol.**
(PDF)

## Acknowledgments

The research team would like to thank the University of Hong Kong and the Prince Phillip Dental Hospital for the crucial supports to this project.

## Author Contributions

**Conceptualization:** Reinhard Chun Wang Chau, Ming Chong, Nate Sing Po Chu, Mohamad Koohi-Moghadam, Richard Tai-Chiu Hsung, Walter Yu Hang Lam.

**Data curation:** Reinhard Chun Wang Chau, Richard Tai-Chiu Hsung, Walter Yu Hang Lam.

**Formal analysis:** Reinhard Chun Wang Chau, Richard Tai-Chiu Hsung, Colman McGrath, Walter Yu Hang Lam.

**Funding acquisition:** Reinhard Chun Wang Chau, Colman McGrath, Walter Yu Hang Lam.

**Investigation:** Reinhard Chun Wang Chau, Ming Chong, Khaing Myat Thu.

**Methodology:** Reinhard Chun Wang Chau, Ming Chong, Richard Tai-Chiu Hsung, Walter Yu Hang Lam.

**Project administration:** Walter Yu Hang Lam.

**Resources:** Reinhard Chun Wang Chau, Ming Chong, Khaing Myat Thu, Nate Sing Po Chu, Mohamad Koohi-Moghadam, Richard Tai-Chiu Hsung, Colman McGrath, Walter Yu Hang Lam.

**Software:** Reinhard Chun Wang Chau, Richard Tai-Chiu Hsung.

**Supervision:** Reinhard Chun Wang Chau, Walter Yu Hang Lam.

**Validation:** Reinhard Chun Wang Chau, Nate Sing Po Chu.

**Visualization:** Reinhard Chun Wang Chau.

**Writing – original draft:** Reinhard Chun Wang Chau.

**Writing – review & editing:** Reinhard Chun Wang Chau, Ming Chong, Khaing Myat Thu, Mohamad Koohi-Moghadam, Richard Tai-Chiu Hsung, Colman McGrath, Walter Yu Hang Lam.

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
