## [Decision Letter · Decision Letter 0]

7 Mar 2022

PONE-D-21-39647

Artificial intelligence-designed single tooth dental prostheses: a protocol

PLOS ONE

Dear Dr. Lam,

Thank you for submitting your manuscript to PLOS ONE. After careful consideration, we feel that it has merit but does not fully meet PLOS ONE’s publication criteria as it currently stands. Therefore, we invite you to submit a revised version of the manuscript that addresses the points raised during the review process.

We look forward to receiving your revised manuscript.

Kind regards,

Gaetano Isola, Ph.D.

Academic Editor

PLOS ONE

Journal Requirements:

Additional Editor Comments:

In view of the negative comments raised by the reviewers, the manuscript cannot be accepted in its current form.

Reviewers' comments:

Reviewer's Responses to Questions

**Comments to the Author**

1. Does the manuscript provide a valid rationale for the proposed study, with clearly identified and justified research questions?

Reviewer #1: Partly

Reviewer #2: Partly

Reviewer #3: Yes

Reviewer #4: Yes

2. Is the protocol technically sound and planned in a manner that will lead to a meaningful outcome and allow testing the stated hypotheses?

Reviewer #1: Yes

Reviewer #2: No

Reviewer #3: Yes

Reviewer #4: Yes

3. Is the methodology feasible and described in sufficient detail to allow the work to be replicable?

Reviewer #1: Yes

Reviewer #2: No

Reviewer #3: Yes

Reviewer #4: No

4. Have the authors described where all data underlying the findings will be made available when the study is complete?

Reviewer #1: Yes

Reviewer #2: No

Reviewer #3: Yes

Reviewer #4: No

5. Is the manuscript presented in an intelligible fashion and written in standard English?

Reviewer #1: Yes

Reviewer #2: No

Reviewer #3: Yes

Reviewer #4: Yes

6. Review Comments to the Author

You may also provide optional suggestions and comments to authors that they might find helpful in planning their study.

Reviewer #1: Some recommendations follow after reading the authors' manuscript with great interest.

The title should mention "posterior" or "molar" tooth.

It should also be mentioned what type of study this protocol is.

It could mention that the teeth movement during MIP (due to PDL compression) was not considered.

In the inclusion criteria, it is better to refer to the "functional occlusal units" as defined by Kyser instead of the occluding pairs.

Static occlusal morphology is not in a satisfactory term. Static dental occlusion could be correct. Consider mentioning it as MIP or centric occlusion.

Consideri citing the following references to support your work:

- Piedra-Cascón W, Hsu VT, Revilla-León M. Facially driven digital diagnostic waxing: New software features to simulate and define restorative outcomes. Current Oral Health Reports. 2019;6(4):284-94.

- Revilla-León M, Gómez-Polo M, Vyas S, Barmak BA, Gallucci GO, Att W, Özcan M, Krishnamurthy VR. Artificial intelligence models for tooth-supported fixed and removable prosthodontics: A systematic review. J Prosthet Dent. 2021 Jul 16:S0022-3913(21)00309-7.

- Revilla-León M, Gómez-Polo M, Vyas S, Barmak BA, Galluci GO, Att W, Krishnamurthy VR. Artificial intelligence applications in implant dentistry: A systematic review. J Prosthet Dent. 2021 Jun 15:S0022-3913(21)00272-9.

- Hasanzade M, Aminikhah M, Afrashtehfar KI, Alikhasi M. Marginal and internal adaptation of single crowns and fixed dental prostheses by using digital and conventional workflows: A systematic review and meta-analysis. J Prosthet Dent. 2021 Sep;126(3):360-368.

- Afrashtehfar KI, Ahmadi M, Emami E, Abi-Nader S, Tamimi F. Failure of single-unit restorations on root filled posterior teeth: a systematic review. Int Endod J. 2017 Oct;50(10):951-966.

Reviewer #2: The idea of the research has potential to lead to a new era of digital dentistry applications. however the described methods are weak and seems insufficient to be further evaluated

Reviewer #3: Well written protocol that will be assessed the most important topic. Methodology is reasonable. But authors should clarify how they determine the sample size.

Reviewer #4: Dear Authors,

The aim of this protocol to develop AI model to design dental prosthesis. While the protocol is helpful for future dentists, some concerns were raised. Revise the manuscript by following comments.

Major points

Figure 2 is not readable.

Some of details are lacking. Specification of high-performance computer, voxel size, and image size. I also could not find such information from supplemental.

Best fit superimposition is a function implemented on the software. But, it was black box.

Any references for VoxNex, GVCNN, PointNet++, and FusionNEt?

Minor points

3D should be spelled out at the first use.

As for the sample size, 1000 cases? or 1000 images?

No figure legend? Even in the supplemental.

7. PLOS authors have the option to publish the peer review history of their article (what does this mean?). If published, this will include your full peer review and any attached files.

Reviewer #1: No

Reviewer #2: No

Reviewer #3: No

Reviewer #4: No

---

## [Author Response · Author response to Decision Letter 0]

26 Mar 2022

Reviewer 1

Some recommendations follow after reading the authors' manuscript with great interest.

Response: 

We are glad that reviewer 1 found this manuscript with great interest. Thank you for your comments and suggestions. We have made the following changes accordingly:

The title should mention "posterior" or "molar" tooth.

It should also be mentioned what type of study this protocol is.

- We have amended the title to be “Artificial Intelligence-designed Single Molar Dental Prostheses: A Protocol of Prospective Experimental Study” and the information of the study design is added to the content.

In the inclusion criteria, it is better to refer to the "functional occlusal units" as defined by Kyser instead of the occluding pairs.

- We corrected wording of “functional occlusal units” in place of “occluding pairs”.

“Subjects with more than 12 functional occlusal units and … …”

It could mention that the teeth movement during MIP (due to PDL compression) was not considered.

Static occlusal morphology is not in a satisfactory term. Static dental occlusion could be correct. Consider mentioning it as MIP or centric occlusion.

- Thank you very much for this comment. We agree with this comment and modified the session “Outcome measurement” accordingly. We have removed “static” and “dynamic” that may cause confusion. We use “Functional occlusal morphology” and this in line with the “functional occlusal units” as in previous comment.

- We will consider and mention the tooth movement during MIP due to PDL compression in our future clinical studies. Thank you for this invaluable comment.

“The limitation of this AI system includes teeth movement within periodontal ligament was not considered.”

Consideri citing the following references to support your work:

- Piedra-Cascón W, Hsu VT, Revilla-León M. Facially driven digital diagnostic waxing: New software features to simulate and define restorative outcomes. Current Oral Health Reports. 2019;6(4):284-94.

- Revilla-León M, Gómez-Polo M, Vyas S, Barmak BA, Gallucci GO, Att W, Özcan M, Krishnamurthy VR. Artificial intelligence models for tooth-supported fixed and removable prosthodontics: A systematic review. J Prosthet Dent. 2021 Jul 16:S0022-3913(21)00309-7.

- Revilla-León M, Gómez-Polo M, Vyas S, Barmak BA, Galluci GO, Att W, Krishnamurthy VR. Artificial intelligence applications in implant dentistry: A systematic review. J Prosthet Dent. 2021 Jun 15:S0022-3913(21)00272-9.

- Hasanzade M, Aminikhah M, Afrashtehfar KI, Alikhasi M. Marginal and internal adaptation of single crowns and fixed dental prostheses by using digital and conventional workflows: A systematic review and meta-analysis. J Prosthet Dent. 2021 Sep;126(3):360-368.

- Afrashtehfar KI, Ahmadi M, Emami E, Abi-Nader S, Tamimi F. Failure of single-unit restorations on root filled posterior teeth: a systematic review. Int Endod J. 2017 Oct;50(10):951-966.

- We have read your recommended papers and enriched our content by adding these classical papers as the references (7, 15-17) correspondingly.

Reviewer 2

The idea of the research has potential to lead to a new era of digital dentistry applications. however the described methods are weak and seems insufficient to be further evaluated

Response: 

We would like to offer our sincere apology for not providing enough information for your evaluation. The manuscript has been revised extensively to provide more details of the study design. Research rationale is supplemented and clarified in Background section, and further details of the study protocol along with new references are added to Material and Method section, including deep-learning methods, outcome measurements and sample size determination. Figures are updated accordingly. We have modified such points to improve the protocol quality and readability. We hope that modifications fit with reviewer's concern for an extent.

Reviewer 3

Well written protocol that will be assessed the most important topic. Methodology is reasonable. But authors should clarify how they determine the sample size.

Response: 

Thank you for your comments and suggestions. We have modified “Sample Size Determination” section with details on deciding the sample size of our training data:

“Each sample, training or validation set, of this study refers to the digitalized tooth model(s) collected from one participant. In a systematic review of the sample size determination for medical AI studies, only 4 studies provided rationale of sample size calculation (ref 37), and only one study is about CNN and 3D images (ref 38). The recommended sample size for CNN in the medical imaging 3D computed tomography has been found to be 1000. From its Receiver Operating Characteristic (ROC) curve, the accuracy of AI system starts to reach plateau when the sample size reaches 200 and become saturated when reach 1000. Therefore, 200 subjects will be recruited as the minimal working sample size for training the AI system. More subjects that are eligible will be recruited for AI training if possible. Additional 50 subjects will be recruited for the validation sets.”

Ref 37: Balki I, Amirabadi A, Levman J, Martel AL, Emersic Z, Meden B, et al. Sample-Size Determination Methodologies for Machine Learning in Medical Imaging Research: A Systematic Review. Canadian Association of Radiologists Journal. 2019;70(4):344-53 

Ref 38: Junghwan Cho KL, Ellie Shin, Garry Choy, and Synho Do. HOW MUCH DATA IS NEEDED TO TRAIN A MEDICALIMAGE DEEP LEARNING SYSTEM TO ACHIEVE NECES-SARY HIGH ACCURACY? arXiv. 2016

Reviewer 4

The aim of this protocol to develop AI model to design dental prosthesis. While the protocol is helpful for future dentists, some concerns were raised. Revise the manuscript by following comments.

Response: 

Major points

Figure 2 is not readable.

- Thank you for your advice. We have scaled up the resolution of figure 2.

Some of details are lacking. Specification of high-performance computer, voxel size, and image size. I also could not find such information from supplemental.

- The specification of university supercomputer is added, with the following content: 

“Data will be computed by the High-Performance Computer provided by the Information Technology Services, The University of Hong Kong, which is capable of conducting data intensive and accelerated computing using a high-performance computing cluster with an aggregating processing power of 804 TFlop/s”. A figure of detailed specification is also added as Fig 3. 

- The voxel size of 3D image (64x64x64) and the resolution of image view (1920x1200) pixels are added.

Best fit superimposition is a function implemented on the software. But, it was black box.

- Thank you very much for your comment. Superimposition is the most commonly used method in the computer image research and digital dentistry. We agree that it was a black box and there may be error associated with it. We have proposed the use of universal coordinate to address the problem of best-fit superimposition. More studies are needed before we can use this universal coordinate in this AI study.

Ref: Pan, Y., Heng, C., Wu, Z. J., Tam, J., Hsung, R. T., Pow, E. H., & Lam, W. Y. (2022). Comparison of the virtual techniques in registering single implant position with a universal-coordinate system: An in vitro study. Journal of Dentistry, 117, 103925.

Any references for VoxNex, GVCNN, PointNet++, and FusionNEt?

- Thank you for your comment. References for the 4 mentioned algorithms have been added in the revised manuscript. 

- Ref 31: Maturana D, Scherer S, editors. VoxNet: A 3D Convolutional Neural Network for real-time object recognition. 2015 IEEE/RSJ International Conference on Intelligent Robots and Systems (IROS); 2015 28 Sept.-2 Oct. 2015

Ref 32: Feng Y, Zhang Z, Zhao X, Ji R, Gao Y, editors. GVCNN: Group-View Convolutional Neural Networks for 3D Shape Recognition. 2018 IEEE/CVF Conference on Computer Vision and Pattern Recognition; 2018 18-23 June 2018

Ref 33: Charles R. Qi LY, Hao Su, Leonidas J. Guibas. PointNet++: Deep Hierarchical Feature Learning onPoint Sets in a Metric Space arXiv. 2017

Ref 34: Quan TM, Hildebrand DGC, Jeong W-K. FusionNet: A Deep Fully Residual Convolutional Neural Network for Image Segmentation in Connectomics. Frontiers in Computer Science. 2021;3

Minor points

3D should be spelled out at the first use.

- The first mention of “3D” is changed to “three-dimensional (3D)”. Thank you for your comment.

As for the sample size, 1000 cases? or 1000 images?

- Each sample or training set of this study refers to the digitalized maxillary tooth models collected from one subject. This statement is now implemented in the Sample size session.

“Each sample, training or validation set, of this study refers to the digitalized tooth model(s) collected from one participant.”

No figure legend? Even in the supplemental.

Thank you very much for you comment. Our apology for the inconvenience. We have prepared a word file with the figures and legends.

Summary

This manuscript is now revised according to reviewers’ precious comments and thanks for your suggestions. We look forward to hearing from you regarding our submission and we would be glad to respond to any further questions and comments that you may have.

---

## [Decision Letter · Decision Letter 1]

3 May 2022

Artificial intelligence-designed single molar dental prostheses: a protocol of prospective experimental study

PONE-D-21-39647R1

Dear Dr. Lam,

We’re pleased to inform you that your manuscript has been judged scientifically suitable for publication and will be formally accepted for publication once it meets all outstanding technical requirements.

Kind regards,

Gaetano Isola, Ph.D.

Academic Editor

PLOS ONE

Additional Editor Comments (optional):

Reviewers' comments:

Reviewer's Responses to Questions

**Comments to the Author**

1. Does the manuscript provide a valid rationale for the proposed study, with clearly identified and justified research questions?

Reviewer #1: Yes

Reviewer #3: Yes

Reviewer #4: Yes

2. Is the protocol technically sound and planned in a manner that will lead to a meaningful outcome and allow testing the stated hypotheses?

Reviewer #1: Partly

Reviewer #3: Yes

Reviewer #4: Yes

3. Is the methodology feasible and described in sufficient detail to allow the work to be replicable?

Reviewer #1: Yes

Reviewer #3: Yes

Reviewer #4: Yes

4. Have the authors described where all data underlying the findings will be made available when the study is complete?

Reviewer #1: Yes

Reviewer #3: Yes

Reviewer #4: Yes

5. Is the manuscript presented in an intelligible fashion and written in standard English?

Reviewer #1: Yes

Reviewer #3: Yes

Reviewer #4: Yes

6. Review Comments to the Author

You may also provide optional suggestions and comments to authors that they might find helpful in planning their study.

Reviewer #1: The Reviewer #1 is pleased with the authors’ corrected version of their protocol manuscript.

There are no additional comments.

Reviewer #3: Authors addressed appropriately my comments

Reviewer #4: Dear Authors,

The manuscript was well revised according to the reviewer's comments. Thanks for your effort.

7. PLOS authors have the option to publish the peer review history of their article (what does this mean?). If published, this will include your full peer review and any attached files.

Reviewer #1: **Yes: **Kelvin I. Afrashtehfar

Reviewer #3: No

Reviewer #4: No

---

## [Editor Report · Acceptance letter]

19 May 2022

PONE-D-21-39647R1 

Artificial intelligence-designed single molar dental prostheses: a protocol of prospective experimental study 

Dear Dr. Lam:

I'm pleased to inform you that your manuscript has been deemed suitable for publication in PLOS ONE. Congratulations! Your manuscript is now with our production department. 

Kind regards, 

on behalf of

Prof. Gaetano Isola 

Academic Editor

PLOS ONE